# Thermal and Gluing Properties of Phenol-Based Resin with Lignin for Potential Application in Structural Composites

**DOI:** 10.3390/polym15020357

**Published:** 2023-01-10

**Authors:** Danilo Soares Galdino, Marcel Yuzo Kondo, Victor Almeida De Araujo, Gretta Larisa Aurora Arce Ferrufino, Emerson Faustino, Herisson Ferreira dos Santos, André Luis Christoforo, Carlos Manuel Romero Luna, Cristiane Inácio de Campos

**Affiliations:** 1Department of Mechanical Engineering, São Paulo State University (UNESP), 333 Doutor Ariberto Pereira da Cunha Avenue, Guaratinguetá 12516-410, Brazil; 2Civil Engineering Postgraduate Program, Federal University of São Carlos (UFSCar), 235 km Washington Luís Highway, São Carlos 13565-905, Brazil; 3Science and Engineering Institute, São Paulo State University (UNESP), 519 Geraldo Alckmin Street, Itapeva 18409-010, Brazil; 4Campus of Ariquemes, Federal Institute of Education, Science and Technology of Rondônia (IFRO), Ariquemes 76870-000, Brazil

**Keywords:** lignin-phenol-formaldehyde, bio-based resin, structural composites

## Abstract

Using Kraft lignin, bio-based adhesives have been increasingly studied to replace those petrochemical-based solutions, due to low cost, easy availability and the potential for biodegradability of this biomaterial. In this study, lignin-based phenol-formaldehyde (LPF) resins were synthesized using commercial Eucalypt Kraft Lignin (EKL), purified at 95%, as a phenol substitute in different proportions of 10%, 20%, 30% and 50%. The properties of bio-based phenol formaldehyde (BPF) synthesized resin were compared with phenol-formaldehyde resin (PF) used for control sampling. The results indicated that viscosity, gel time and solid contents increased with the addition of pure EKL. The shear strength test of glue line was studied according to American Society for Testing and Materials (ASTM), and BPF-based results were superior to samples bonded with the PF as a control sample, being suitable for structural purposes. Changes in the curing behavior of different resins were analyzed by Differential Scanning Calorimetry (DSC), and sample comparison indicated that the curing of the LPF resin occurred at lower temperatures than the PF. The addition of EKL in PF reduced its thermal stability compared to traditional resin formulation, resulting in a lower decomposition temperature and a smaller amount of carbonaceous residues.

## 1. Introduction

Phenol-formaldehyde resins (PF) are among the most versatile polymers ever invented, which continue to be widely used by the industry in diversified products. This adhesive is synthesized from the polycondensation of phenol and formaldehyde in a process in which two possible adhesives are extracted: resol-type resins, using a molar excess of formaldehyde over phenol, under basic conditions; and novolac-type resins, using a molar excess of phenol over formaldehyde, under acidic conditions [1].

The choice of the better resin type depends on the final application of lignocellulosic composite, as bio-based compounds can be developed under different compositions and manufactures towards multiple uses. In practice, this composition changes according to the proportions of materials present in the formulation.

In timber and construction industries, resol resins are used in the manufacture of oriented strand board (OSB) and plywood [2,3], which are very suitable for outdoor applications due to their excellent mechanical properties, low cost, high thermal stability, and water resistance [4,5]. These composites have been frequently applied in industrialized construction systems such as light-woodframe and steelframe, whose structural uses in both technologies include wall sealing, ceiling, subflooring, and roofing base [6].

Both phenol and formaldehyde are toxic substances present in the phenol adhesives due to the damage caused to the environment [7], which demand sustainable raw materials and modifications in a synthetic route to improve mechanical properties. Furthermore, these substances are derived from the oil industry, whose resources are deeply marked by unstable costs and non-sustainable origins. Thus, it is necessary to find a substitute for this traditional resin through an ecologically alternative based on safe use for humans, lower costs without the dependence on volatile prices of this commodity and, above all, with similar mechanical properties.

The need for “green chemistry” has encouraged the recent interest for naturally sourced substitutes, which are readily available from renewable raw materials in large quantities and with lower costs [8]. In this aspect, lignin is considered a promising source of new reagent from biomass, carbon neutral and inedible [9]. After cellulose, lignin is the most abundant natural biopolymer found, being present in all plants in considerable volumes, ranging from 15 to 25% of the dry weight of woody plants [10].

Lignin is a complex and amorphous bio-based polyphenol, which is synthesized from the polymerization of the following phenyl-propane monomers: coumarin, synapyl, and coniferyl alcohols [11]. Its structure is rich in aromatics, which makes it a very appealing sustainable alternative to potentially replace petroleum-derived phenol.

Over 150 million tons of lignin is biosynthesized each year by terrestrial plants, and about one third of this bioproduction has been converted into a by-product of the pulp and paper, where recently it was still considered an industrial waste [8,12]. However, the extraction process represents the key point for the use of lignin in industrial applications, as their main properties can vary according to the raw material and pulping conditions, especially in reason of the number of reactive groups and their molecular weight [13,14].

Kraft lignin, which is produced in wood pulping as a by-product named black liquor, has been increasingly studied in the production of resins [9,14,15,16]. This residue production has grown at a global level, as the methods of lignin extraction have become more efficient and optimized on a commercial scale in the recent years [16]. Literature has addressed two methods to synthesize lignin-based resins: catch all, a method that utilizes lignin without purification; and the purified method, which includes a purification and a modification by methylolation, phenolation and demethylation [17]. This modification is necessary to improve its reactivity.

Recently, the influence of some modification methods were investigated regarding the adhesion properties of lignin-phenol-formaldehyde resin, and this study concluded that, regardless the modification method, the use of lignin to replace phenol effectively improves the gluing performance. Lignin was extracted in laboratory using a synthesis with duration of 240 min [18]. Despite the knowledge on lignin on different aspects related to potential of use, structure and modification, further studies are required to understand practical issues associated to the intrinsic variability of lignin from different sources or processes, and the respective improvement processes for industry [14]. It is known that the properties of these resins are highly affected by the molar ratios of lignin, phenol and formaldehyde in the utilization, synthesis temperature, synthesis time and pH of the mixture [19,20,21,22]. Literature addressed that there are 22 lignin producers, 80% of which are focused on lignosulfonate lignin [16]. This reason justifies greater attention to the studies related to lignin from other processes, and/or extracted from black liquor and purified in laboratory conditions [6,12,18,20,23], as opposed to lignin from the Kraft pulp industry, whose extraction methods have become more efficient and commercial [16]. Thereby, it should be noted that different lignin materials present different physical and chemical properties [22].

Due to this raised scenario, the motivation of this research aimed to investigate the use of high purity *Eucalyptus* spp. lignin from Kraft pulping in the production of phenolic resin to be used in the manufacture of wood-based panels for structural purposes, whose synthesis conditions were not optimized so far. The goal was to improve the production efficiency using shorter synthesis times; that is, 120 min instead of 240 min. The lignin was used as a substitute for phenol in different substitution ratios (10, 20, 30 and 50%) in the synthesis of resol resins. In addition to the characterization of the adhesive and shear strength test in the glue line, thermal performances of these materials are also investigated.

This paper is organized into the following sections: materials and methods with the respective materials, lignin phenolation, syntheses of the phenol-formaldehyde synthesis and lignin-phenol-formaldehyde, resin properties, mechanical test, thermal test, results on physicochemical and mechanical properties, thermal analyses (differential scanning calorimetry and thermal gravimetric analysis), discussion and conclusion.

## 2. Materials and Methods

### 2.1. Materials

This work utilized Kraft Lignin from an anonymous company oriented to produce cellulosic pulp from eucalypt wood, which is located at São Paulo state, Brazil. Thus, lignin in powder form was precipitated from the black liquor by the company through pH modification and purified condition at 95%. Methanol (99.7%), sodium hydroxide, NaOH (60%), formaldehyde (37%) and phenol (99.5%) reagents were procured from the “Nox Solutions Industria de Produtos Químicos”, a Brazilian chemical company.

### 2.2. Synthesis of the Phenol-Formaldehyde (PF)

Phenol-formaldehyde resin (PF) was prepared by the reaction of phenol (P) and formaldehyde (F) using a 1:2 molar ratio. 60% aqueous NaOH solution was utilized as a catalyst and 50% aqueous methanol solution as a solvent. The synthesis was prepared in a reactor equipped with a stirrer, thermometer and a reflux condenser.

Studies [19,20] utilized as a reference for the initial configurations of molar ratio and synthesis conditions. The reagents (50 g phenol (99.5%), 90 g formaldehyde at 37%, 20 g aquous solution of methanol at 50%) were inserted in the reactor and mechanically stirred at 200 RPM and heating at 80 °C for 2 h. Three 2.4 g NaOH charges were added, being the first one in the initial moment, and the others after 10 and 20 min.

### 2.3. Lignin Phenolation

Kraft lignin is hydrophobic and it is not an active chemical compound, although the same could be modified to improve its reactivity [11]. The purpose of this method aimed to increase the reactivity as well as to provide/facilitate the adhesive synthesis and polymerization reaction.

Lignin and phenol (p.a.) were inserted into a flask, in amounts (by mass) using four proportions (10%, 20%, 30%, and 50% EKL). Phenol-lignin mixture was kept into heated bath at 40 °C for 1 h, in a process to obtain a homogeneous mass (phenol-lignin) for the synthesis of phenolic resin.

### 2.4. Synthesis of the Lignin-Phenol-Formaldehyde (LPF)

The condition and synthesis equipment of LPF were similar to PF resin. Phenol modified by phenolation, formaldehyde and methanol were also heated to 80 °C and kept at this temperature for 2 h, after the gradual addition of sodium hydroxide (60%). Methanol (20 g; 50% *v*/*v*) and sodium hydroxide (7.2 g; 60%) were used in all synthesized adhesives.

Table 1 represents the amounts of lignin, phenol, formaldehyde utilized in the different formulations of LPF resins. The partial replacement of the phenol mass by Kraft lignin considered the four different conditions: 10%, 20%, 30%, and 50% designed by the following resin nomenclature: LPF10, LPF20, LPF30 and LPF50, respectively.

### 2.5. Resin Properties

#### 2.5.1. Viscosity

A Ford cup with a 4 mm orifice was used in the viscosity testing according to ASTM D1200-2005 (Standard Test Method for Viscosity by Ford Viscosity Cup).

#### 2.5.2. pH

The potential of hydrogen (pH) values of all resin samples were obtained by direct measurements using a pHmeter device (Digimed DM22).

#### 2.5.3. Solids Content

The quantification of solids was determined by evaporating resin samples at 120 °C for 2 h, according to ASTM standard D4426–01 (2006). Contents were calculated in percentage, using the ratio between the final and the initial mass.

#### 2.5.4. Gel Time

For each sample, 5 g of adhesive were used to determine their respective gel times. Each adhesive was placed in a test tube, and immersed in glycerin at 130 °C, being constantly moved through a glass rod to reach the gel phase in each evaluated sample. Time elapsed in this evaluation was recorded and corresponded to the gel time value.

### 2.6. Mechanical Test

In the evaluation of strength test of glue line, the shear test of the glued joints was carried out, following recommendations contained in the standard ASTM D-2339 (2020). Glued joints were produced using two Pinewood strips (40 cm × 10 cm × 0.5 cm each), which were dried up to be stabilized at ±10% moisture content. The synthesized adhesives were used in a 180 g/m^2^ weight, in a double line, being manually spread with the support of a spatula. Strips were pre-pressed for 2 min and, sequentially, they were pressed using a hydraulic press according to the following parameters: 0.7 MPa pressure, 180 °C temperature, and 6 min curing time. Subsequently, 8.26 cm × 2.54 cm specimens for each treatment studied were made.

As specified by the aforementioned standard document, 30 specimens were submitted to the shear stress test to evaluate the strength in glue line though a universal testing machine (Figure 1). Loading speed used was compatible with the adopted standard (600 to 1000 lb/min).

### 2.7. Thermal Tests

#### 2.7.1. Differential Scanning Calorimetry (DSC)

The analysis was carried out with a specific instrumentation (Perkin Elmer Pyris 1), where the studied samples were heated from 30 to 300 °C, at a heating rate of 10 °C min^−1^.

#### 2.7.2. Thermal Gravimetric Analysis (TGA)

Lastly, thermal stability of samples was investigated through a thermogravimetric analyzer (Perkin Elmer TGA 7). The recorded scans ranged from 30 to 1000 °C with a heating rate of 10 °C/min under nitrogen atmosphere.

### 2.8. Fourier Transform Infrared Spectrometry (FTIR)

The synthesized resin samples were analyzed by FT-IR, using a PerkinElmer spectrophotometer, model Spectrum 100, in the spectral range of 4000 to 700 cm^−1^, with a total of 8 scans.

### 2.9. Statistical Analysis

A hypothesis test was performed using the Analysis of Variance (ANOVA) to verify whether there were significant differences between the means. The Tukey Test was used at the 5% nominal significance level of probability.

## 3. Results

### 3.1. Physicochemical Properties

The physical and chemical properties of synthesized resins were described in Table 2, both for traditional phenol-formaldehyde adhesive (PF) and four different formulations of lignin-phenol-formaldehyde (LPF) with 10, 20, 30 and 50% conditions, as specified by Table 1. The pH increased as the replacement percentage was increase. LPF50 reached the highest value (11.64). Resins with a strong alkaline condition can result in the degradation of wood constituents and the corrosion of metal fasteners. Thus, aiming the final utilization of panels, the study prioritized a resin with pH limit of 12.

The highest viscosity values were for the adhesives with the highest replacement percentages. The increase in the viscosity of the PF resin by the addition of wood lignin is associated with physical effects, such as an increase in the solids content, and chemical reasons, such as differences in the molecular weights of the lignin, its ramifications and available reactive sites [22].

With regards to solid content, resins presented results of about 50%, with the exception of LPF10. The resin with lower solid content had faster gelatinization (LPF10: 46.35% and 219 s), while resin with greater solid content has a slower gelatinization (LPF20: 52.16% and 354 s). This difference may be associated to differences that occurred in the reticulation of these adhesives. There was an evaporation of unreacted formaldehyde for resins with lower solid contents, in addition to the water formed in the reaction. There is a study indicating that, compared to synthetic polymers, resins with lignin have a longer gel time due to a lower value of free formaldehyde [24].

### 3.2. Mechanical Properties

Figure 2 presents the average values for the determination of the shear strength in the glue line. The results showed that LPF resins with phenol substitution above 20% showed a superior performance than the traditional PF resin, being the higher value for LPF30 (3.41 MPa).

LPF10 and LPF50 resins did not meet the criteria established by the ASTM D-2559 and ASTM D-5751 standard document, which prescribes specifications for the structural and non-structural uses of glued-timber components. Tested samples shall reach at least 75 and 60% in the wood rupture (Figure 2), respectively. PF and LPF30 met the industrial requirements for structural purposes.

Figure 3 shows the characteristics of resins with greater strength values. Based on these graphs, the parameters defined to obtain an efficient adhesion were: solid content over 48%, pH over 11, and gel time over 300 s.

The projected values indicate that a resin with viscosity around 400 cP would result in even greater resistance in the glue line, indicating that this is a parameter to be optimized [19], with the exception of LPF50. Despite the fact that LPF50 has reached the adequate viscosity, it did not adhere satisfactorily (Figure 3), as it has bonding failures.

### 3.3. Thermal Analyses

#### 3.3.1. Differential Scanning Calorimetry (DSC)

The calorimetry analysis diagrams are illustrated in Figure 4 for different adhesive formulations. According to Khan et al. [20], the endothermic curve standards are a consequence of curing in adhesives. The peak temperature (T_peak_), initial temperature (T_onset_), ΔT (T_peak_ − T_onset_) and enthalpy (ΔH) are presented in Table 3 for the resins under study. The endothermy of LPF resins appeared in the lower temperatures, having greater enthalpies, and progressing in a narrower temperature range. Lower ΔT values indicate that the LPF resin condenses faster compared to traditional PF.

Different groups indicate that LPF resin is modified at lower temperatures and is faster than a PF resin [25,26]. These studies verified that lignin reactivity is essentially influenced by the phenolic hydroxyl number; that is, the greater the presence of this group, the greater the reactivity of lignin in relation to formaldehyde when lignin is used for the formation of phenolic resin [26]. The high hydroxyl content of Kraft lignin is due to the method of pulping process and the raw material.

#### 3.3.2. Thermal Gravimetric Analysis (TGA)

The results of TGA for resins are presented in Figure 5. LPF resins are less stable in the initial stages (up to 150 °C), but at higher temperatures LPF and PF are equivalent. As the resins did not undergo a rotary-evaporation process, a marked mass loss (50%) was observed in the initial phase (up to 150 °C) attributed to the evaporation of the most volatile components (water, methanol), followed to a lesser extent by the elimination of the formaldehyde that remained free in the resin [27]. LPF50 resin showed less stability and a different decomposition. This difference was also observed in other studies such as [26].

Mass loss in the initial stages can also be associated with the breakdown of lignin side chains [25] and between 120–240 °C losses resulting from post-curing resin reactions [20]. After this event, these resins started to decompose more slowly, until at 500 °C a second mass loss occurs due to the breaking of methylene bonds [25]. Above 600 °C, few differences between the resins are observed, because in this range, aromatic structures present in all samples began to degrade, leading to the formation of a carbon residue [10].

The lower generation of carbon from LPF resins (Table 4) tends to worsen the fire resistance properties of the material during burning. On the other hand, the higher temperatures observed at the beginning of decomposition (T5%, T10% and T30%) results in a slight reduction to ignition of LPF resins. Similar conclusions were found by Khan et. al. [20] in their studies replacing phenol with lignin from eucalyptus bark.

Analyzing the DTG curves (Figure 6), it can be concluded that the decomposition of the LPF resins can be divided into four phases: from 0 to 200 °C; from 200 °C to 400 °C; from 400 °C to 600 °C; and above 600 °C. Most of the thermal events occur in the first stage and explain the lower stability of the LPF resin compared to PF.

The displacement of the secondary peak at 150 °C is observed at this stage, which is characteristic of the PF resin to lower temperatures as phenol is replaced by lignin. Likewise, the formation of small peaks is observed in the second stage at temperatures of 250 °C and 350 °C in adhesives with lignin as the peak decreases at 550 °C.

### 3.4. FTIR Analysis

The FTIR spectra of the phenol-formaldehyde (PF) and lignin-phenol-formaldehyde (LPF) resins are demonstrated by Figure 7.

It is observed that, with the exception of the LPF50 resin (50% lignin), the spectra of the other resins show similar peak arrangements (Figure 7), which indicates structural similarity of the resins up to the percentage of 30% of replacement.

The broad absorption bands around 3350 cm^−1^ correspond to –OH groups resulting from polymeric association [28], while the absorption bands at 1635 and 1445 cm^−1^ were related to vibrations of the aromatic ring and the phenyl skeleton-propane [4]. The band at 1244 cm^−1^ was attributed to the presence of C–O stretching vibration of the phenolic hydroxyl groups, while the C–O stretching vibration of aliphatic C–O (Ar), aliphatic C–OH and methylol C–OH was represented by the band at 1015 cm^−1^ [23].

The band at 1115 cm^−1^ increases as the band at 1150 cm^−1^ decreases as lignin is added (Figure 7), revealing that there were more C–O stretching vibrations of the phenolic hydroxyl groups in the PF resin than in the LPF resins [4]. The peak at 1115 cm^−1^ and the new peak that appears in the 1380 band in the LPF50 resin is attributed to the COO and COH of the syringyl units (lignin precursor unit) [20]. These events confirm that lignin was successfully introduced into the phenolic crosslinking system.

## 4. Discussion

The complete replacement of phenol by lignin is still the subject of ongoing research. Thus, the limited use of lignin in resins is generally attributed to its low reactivity with formaldehyde, which can mainly be attributed to its non-uniform structure. In addition to its degree of purity, the raw material used, the pulping conditions and the isolation processes are decisive factors in the choice of lignin, as already reported.

The performance of LPF resins prepared with Kraft lignin were comparable with the performance of adhesives synthesized in previous research using other sources of lignin and longer synthesis times [20,26,28]. This reduction was possible due to the high purity levels and homogeneity of the Kraft lignin obtained in industrial processes, which resulted in a shorter condensation process time during synthesis [29]. Despite the higher enthalpies, the curing of these resins occurred at lower temperatures, which is favorable to the industry.

The high molecular weight and the presence of the aromatic ring in the lignin structure is seen as a difficulty in condensation with phenol and formaldehyde [29], resulting in a decrease in the resistance of resins with lignin at high temperatures [30]. Recent studies, however, confirm that these properties can be improved through crosslinking potentiating agents, which may even be of renewable origin such as furfural [31]. There are indications that its use stimulates the formation of a denser network structure, more resistant to high temperature and water absorption [32,33], characteristics that are desirable for structural adhesives for wood panels.

Another promising method to increase the applicability of lignin to generate value-added products is the conversion of lignin into smaller fragments through depolymerization (cracking). The goal of lignin depolymerization shall convert the complex structure of lignin into fragments that exhibit greater similarity to the chemical structure of phenol than lignin. This property makes depolymerized lignin an extremely interesting alternative to phenol by allowing even higher substitution levels as a result of increased reactivity [24].

Through the valorization of lignin by the wood panel industry in obtaining a resin with a higher content of biological solids generated as residues or by-products by another sector (e.g., the pulp industry), aligns with the principles of the more efficient activities that shall compose the synergies towards bioeconomy as suggested by [34].

## 5. Conclusions

This paper successfully investigated the viability of high purity Kraft lignin made industrially as a substitute for phenol in the lignin-phenol-formaldehyde formulation.

Thus, positive results were observed and therefore the following conclusions were drawn:Lignin-based resins can be synthesized in shorter periods such as 120 min.The physical tests of the four resin formulations indicated that the addition of phenol-lignin tends to influence in a more pronounced way the characteristics of the resins from the substitution percentage of 20%.Differential Scanning Calorimetry analysis showed that the curing of lignin-based resins occurs at lower temperatures with greater enthalpies compared to PF resin.Shear strength in the glue line of LPF resins is compatible with the PF resin in a replacement percentage of 30%, a condition considered ideal as it meets the specifications for structural use.Considering the overall performance in thermal tests, the replacement of phenol by lignin results in a slight decrease in resistance to high temperatures, being compensated by the reduction in ignition of LPF resins in the initial stages.For future works, we suggest the performance of studies towards the increase of substitution levels and the degree of reticulation of resins. In addition, there is the possibility to investigate the behavior of structural panels glued with LPF resins under the effective simulation of fire tests.

## Figures and Tables

**Figure 1 polymers-15-00357-f001:**
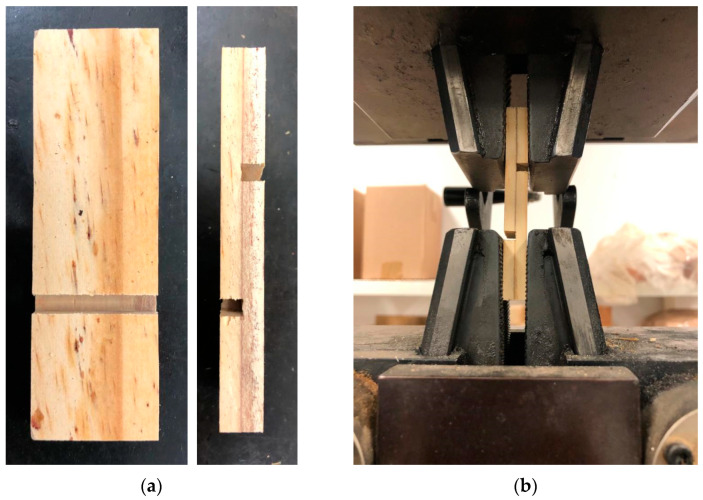
Shear strength test in the glue line: (**a**) specimens, and (**b**) test in operation.

**Figure 2 polymers-15-00357-f002:**
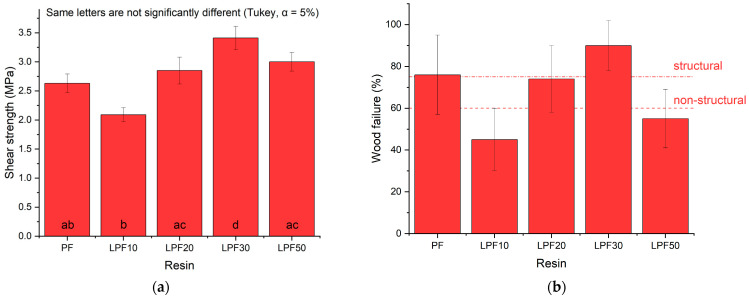
Influence of different resins on: (**a**) tensile shear strength, and (**b**) wood failure.

**Figure 3 polymers-15-00357-f003:**
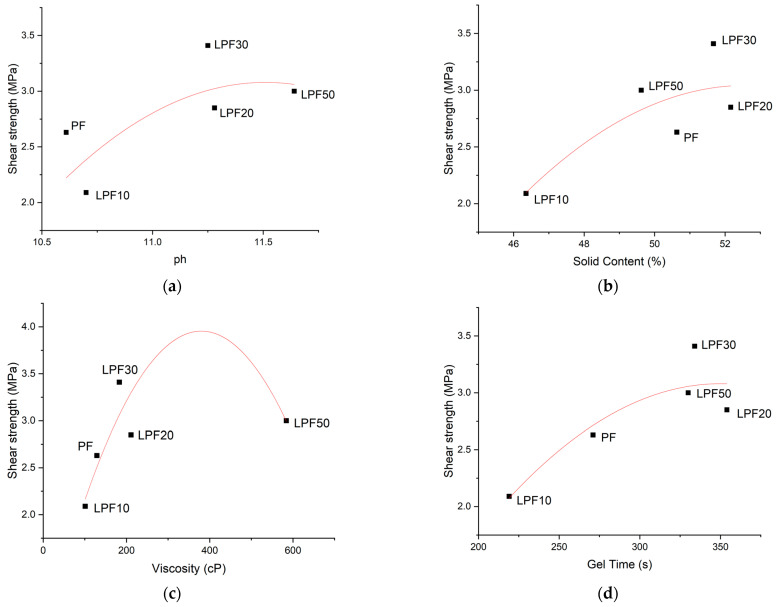
Development of tensile shear strength of different resins as a function of: (**a**) pH, (**b**) solid content, (**c**) viscosity, and (**d**) gel time.

**Figure 4 polymers-15-00357-f004:**
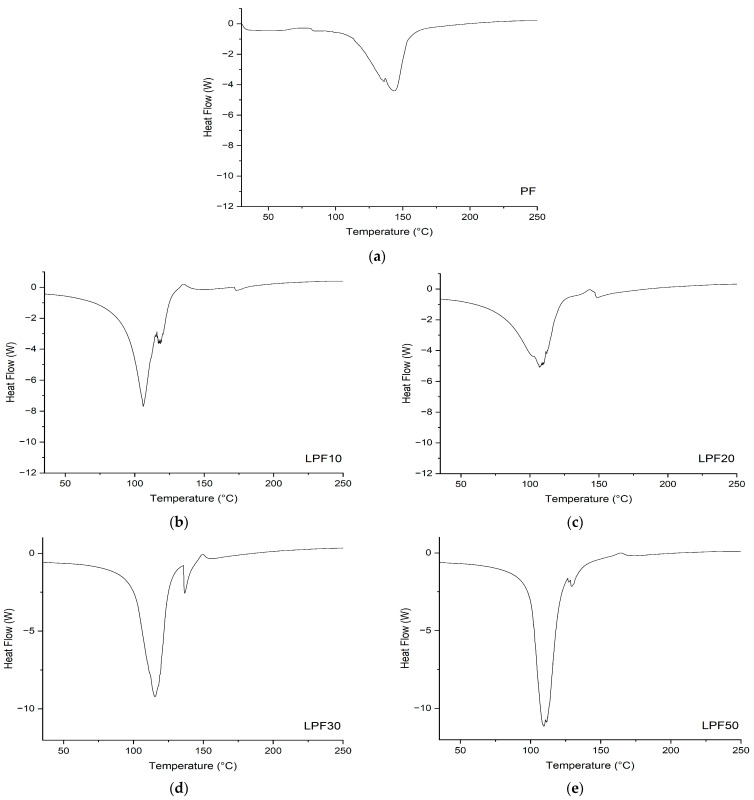
Diagrams of calorimetry analyses (heat flow per temperature) for different resins under study: (**a**) PF, (**b**) LPF10, (**c**) LPF20, (**d**) LPF30 and (**e**) LPF50.

**Figure 5 polymers-15-00357-f005:**
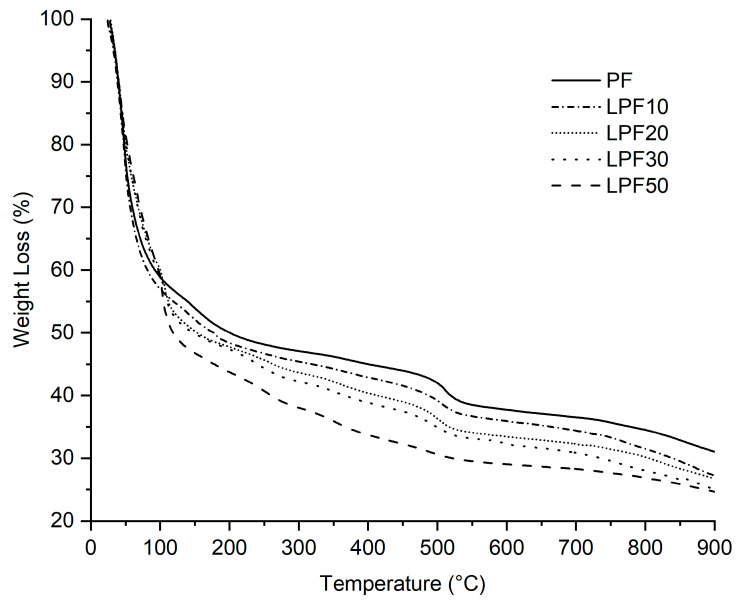
Thermogram results for the different resins.

**Figure 6 polymers-15-00357-f006:**
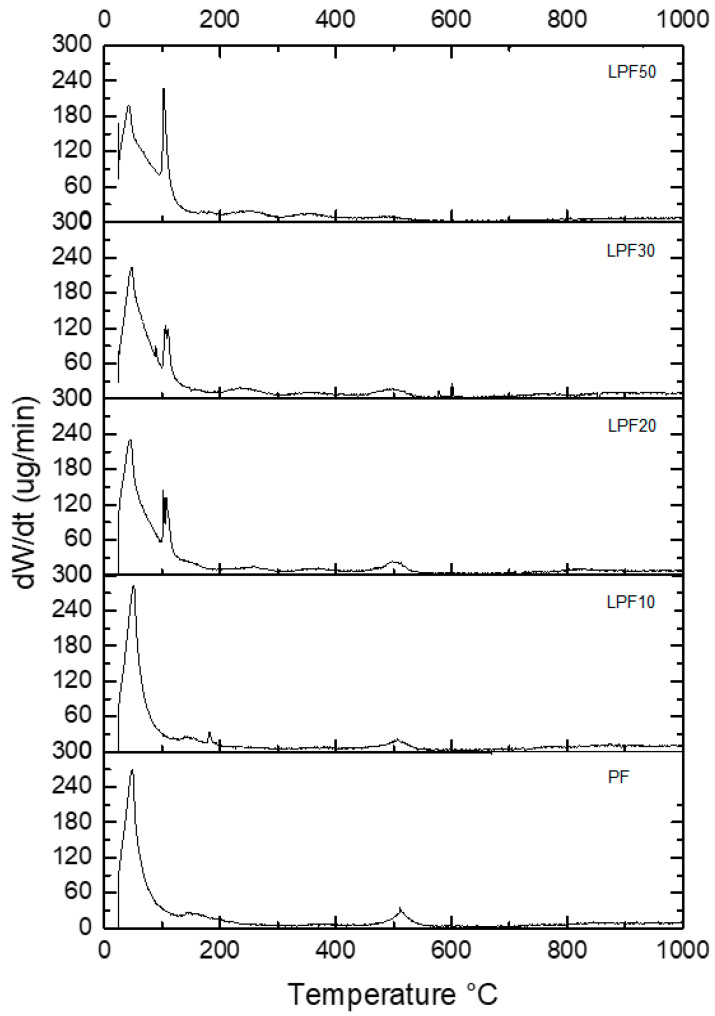
DTG curves of different resins.

**Figure 7 polymers-15-00357-f007:**
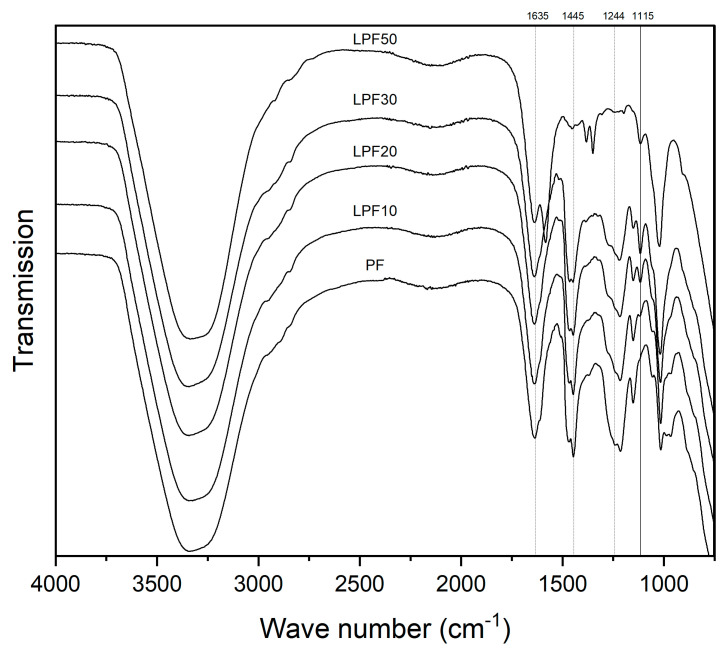
FTIR analysis of different resins.

**Table 1 polymers-15-00357-t001:** Amounts of reagents used in the syntheses of LPF resins.

Resin	Lignin (g)	Phenol (g)	Formaldehyde (g)
PF	0	50	90
LPF10	5	45	90
LPF20	10	40	90
LPF30	15	35	90
LPF50	25	25	90

**Table 2 polymers-15-00357-t002:** Physicochemical properties of studied resins according to observed condition.

Resin	pH	Solids Content (%)	Viscosity (cP)	Gel Time (s)
PF	10.61	50.63 (0.68 ^1^) ^ab^	129 (3.05) ^a^	271 (10.8) ^ab 2^
LPF10	10.70	46.35 (0.09) ^c^	101 (1.47) ^a^	219 (8.82) ^a^
LPF20	11.28	52.16 (0.13) ^d^	211 (1.49) ^b^	354 (17.7) ^c^
LPF30	11.25	51.67 (0.13) ^ad^	183 (3.92) ^b^	334 (21.8) ^bc^
LPF50	11.64	49.62 (0.11) ^b^	584 (23.56) ^c^	330 (22.2) ^bc^

^1^ Standard deviation is in parentheses. ^2^ Same letters (^a^, ^b^, ^c^, and ^d^) in the property results are not significantly different (Tukey, α = 5%).

**Table 3 polymers-15-00357-t003:** Kinetics of thermal curing of resins by DSC.

Resin	T_onset_ ^1^ (°C)	T_Peak_ ^2^ (°C)	ΔT ^3^ (°C)	ΔH ^4^ (J.g^−1^)
PF	122.2	144.1	21.9	623
LPF10	92.2	106.1	13.9	984
LPF20	84.5	107.0	22.5	873
LPF30	98.1	115.6	17.5	1126
LPF50	99.2	109.5	10.3	1211

^1^ T_onset_: initial temperature. ^2^ T_peak_: peak temperature. ^3^ ΔT: temperature difference. ^4^ ΔH: enthalpy.

**Table 4 polymers-15-00357-t004:** Values of TGA analysis for different resins.

Resin	T_5%_ (°C)	T_10%_ (°C)	T_30%_ (°C)	T_60%_ (°C)	Residue_150°C_ (%)	Residue_300°C_ (%)	Residue_600°C_ (%)	Residue_900°C_ (%)
PF	34.6	38.9	59.8	519.9	53.8	47.1	37.7	31.2
LPF10	35.2	38.0	56.6	487.4	52.0	45.4	35.9	27.2
LPF20	35.0	40.7	68.1	414.5	50.2	43.6	33.4	26.8
LPF30	34.2	40.7	68.4	366.3	49.8	42.2	32.4	25.0
LPF50	34.6	40.5	71.8	260.1	46.7	38.0	29.0	24.6

## Data Availability

The data presented in this study are available upon request from the corresponding author.

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
