# Peer review of "Thermal and Gluing Properties of Phenol-Based Resin with Lignin for Potential Application in Structural Composites"

_polymers, 2023, doi:10.3390/polym15020357_

Round 1

Reviewer 1 Report

This work is devoted to the modification of phenol-formaldehyde resins by adding kraft lignin. The topic of modification of synthetic resins and utilization of lignin is relevant, which is reflected in a large number of works on this topic.

The authors conducted a systematic study and obtained interesting results.

However, there are serious comments on the quality of work:

Line 22, 25 BPF?

2.1 How the "purity" of lignin was determined. What quality of reagents were used in the work?

2.3 A more detailed description of the process and the ratio of reagents is needed.

2.4 Similarly as 2.3.

What do the letters "a b c" mean?

Figure 4. Figures TG and DTG are not signed. DTG curves are usually presented as mass loss rates

How do lignin additives relate to "green chemistry". The binders obtained in the work are still based on phenol and formaldehyde.

Early work investigated the mechanisms of formation of lignin-phenol-formaldehyde bonds. Has research been done in this paper?

An important study is the molecular weight distribution, which is missing in this work.

The authors should emphasize the novelty of the work compared to previous studies.

The list of references is formatted with errors. For example 7.8, 16, 23 etc.

Author Response

This work is devoted to the modification of phenol-formaldehyde resins by adding kraft lignin. The topic of modification of synthetic resins and utilization of lignin is relevant, which is reflected in a large number of works on this topic.

R: We appreciate your review to refine our paper as well as your positive opinion about the relevance of this studied topic.

The authors conducted a systematic study and obtained interesting results.

R: We are pleased with the positive opinion about results.

However, there are serious comments on the quality of work:

Line 22, 25 BPF?

R: Thank you. We inserted the respective term “bio-based phenol formaldehyde” in the first mention.

2.1 How the "purity" of lignin was determined. What quality of reagents were used in the work?

R: As it is a commercial product, the cited purity of lignin (above 95%) is given by product specifications. The product has a USDA biopreferred sealing, which certifies it with the 100% renewable origins. Unfortunately, the company does not allow reverse chemistry to carry out other analyzes.

2.3 A more detailed description of the process and the ratio of reagents is needed.

R:  text was re-written to a better comprehension, and this information was inserted.

2.4 Similarly as 2.3.

R:  text was re-organized, and this section is now 2.4. Table 1 was inserted to clarify the topic.

What do the letters "a b c" mean?

Figure 4. Figures TG and DTG are not signed. DTG curves are usually presented as mass loss rates

R: it was corrected and new graphs were inserted.

How do lignin additives relate to "green chemistry". The binders obtained in the work are still based on phenol and formaldehyde.

R: Synthesized resin did not utilized ligning as a reinforcement material, but it replace phenol by lignin in the formulations. Text was re-written, and table 1 was inserted to clarify it. This replacement aimed to add more sustainability to this product, as the reduction of a non-renewable compound was evidently put into practice. Studies mentioned showed the reduction in the formaldehyde emissions, which will be studied in the sequence of this study for this and other ends. Future suggestions were inserted under this further perspectives.

Early work investigated the mechanisms of formation of lignin-phenol-formaldehyde bonds. Has research been done in this paper?

R: Section 3.4 was inserted and infrared spectra identified the main active groups and those changes in the chemical structures of resins.

An important study is the molecular weight distribution, which is missing in this work.

R: All studies utilized by this research did not carry this specific analysis, which was not used in the presente research due to technical limitations. But, this point was raised as suggestion for future studies.

The authors should emphasize the novelty of the work compared to previous studies.

R: The introduction was rewritten to evince the goals, above all, about the high purity of the lignin in utilization from Kraft process, being a different raw material when compared to other studies. Furthermore, the synthesis time was reduced to evince temporal gains.

The list of references is formatted with errors. For example 7.8, 16, 23 et

R: The list was corrected.

Reviewer 2 Report

This paper is about lignin-based phenol-formaldehyde (LPF) resins made with commercial Eucalypt Kraft Lignin (EKL) as a phenol substitute in different amounts of 10%, 20%, 30%, and 50% for structural wood-based panels. Even though there has been a lot of research on phenolic resins made from lignin, the authors of this study say that it is unique that the lignin used in this study is about 95% pure. This paper makes a scientific contribution by comparing the performance of LPF and conventional phenol-formaldehyde (PF).

After carefully evaluating the manuscript, I concluded that it could not be accepted in its current form and would require significant revisions: The following is a list of my comments:

1) The Mechanical Shear Test

1.1 How many repetitions were used to calculate the average shear strength?

1.2 Explain the type of adhesive failure by displaying images/micrographs of adhesive fracture surfaces.

1.3 Use visual data plots to show the relationship between the samples' viscosity, gel time, solid content, and shear strengths.

2. Infrared spectra of the LPF and PF should be provided so that chemical composition and variation can be studied.

3. Lines 214–218 should be rephrased. Please thoroughly revise the grammar of this manuscript

4. DSC

4.1 The data was misconstrued by the authors, and the DSC results should be re-analyzed. So, the whole discussion is wrong at its core. Please read up on how resole type PF resin behaves in a DSC test. Here are some good examples:

https://doi.org/10.1108/03699421111130432 

https://doi.org/10.1002/app.20374

4.2 Please also give the enthalpy of the reaction.

5. TGA

5.1 Need to explain the strange way that LPF and PF decompose when heated, as shown by the fact that the initial rate of mass loss was sharp but slowed down when about 50% of the mass was still under 100 C. From the pattern of degradation, it is evident that lignin has no effect on the thermal stability of PF. Both LPF and PF degrade similarly. The reduction in residue was related to the amount of lignin replaced in PF.

6. Conclusion

Based on your observations, please conclude what is the best composition (optimized parameter) for the LPF.

Author Response

This paper is about lignin-based phenol-formaldehyde (LPF) resins made with commercial Eucalypt Kraft Lignin (EKL) as a phenol substitute in different amounts of 10%, 20%, 30%, and 50% for structural wood-based panels. Even though there has been a lot of research on phenolic resins made from lignin, the authors of this study say that it is unique that the lignin used in this study is about 95% pure. This paper makes a scientific contribution by comparing the performance of LPF and conventional phenol-formaldehyde (PF).

R: We appreciate your review to refine our paper as well as your positive opinion about the relevance of this studied topic.

After carefully evaluating the manuscript, I concluded that it could not be accepted in its current form and would require significant revisions: The following is a list of my comments:

R: We checked the manuscript using all suggestions to satisfy your expectations.

1) The Mechanical Shear Test

1.1 How many repetitions were used to calculate the average shear strength?

R: As specified by ASTM D-2339(2020), 30 specimens were tested. This information was duly inserted.

1.2 Explain the type of adhesive failure by displaying images/micrographs of adhesive fracture surfaces.

R: As proposed, we carried out the failure analysis using criteria established by the ASTM D-2559(2018), which allowed the conclusion that LPF30 resin may be classified as structural ones, and LPF20 as a non-structural resin. LPF10 and LPF50 did not meet the specifications. Such information was clearly explained in the text.

1.3 Use visual data plots to show the relationship between the samples' viscosity, gel time, solid content, and shear strengths.

R: Graphs were plotted and inserted to evince the relationship among these properties as you can check now.

  1. Infrared spectra of the LPF and PF should be provided so that chemical composition and variation can be studied.

R: As suggested, the analysis of infrared spectra was performed and identified in the text to evince the main chemical composition as you can verify now.

  1. Lines 214–218 should be rephrased. Please thoroughly revise the grammar of this manuscript

R: This part was rewritten as well as there are substantial changes to attend your opinion.

  1. DSC

4.1 The data was misconstrued by the authors, and the DSC results should be re-analyzed. So, the whole discussion is wrong at its core. Please read up on how resole type PF resin behaves in a DSC test. Here are some good examples:

- https://doi.org/10.1108/03699421111130432

- https://doi.org/10.1002/app.20374

R: As suggested, both studies were included. The analyses were carried out using these works and we concluded that LPF resin is cured in lower temperatures with greater enthalpies.

4.2 Please also give the enthalpy of the reaction.

R: We inserted these enthaply values of the reaction of curing.

  1. TGA

5.1 Need to explain the strange way that LPF and PF decompose when heated, as shown by the fact that the initial rate of mass loss was sharp but slowed down when about 50% of the mass was still under 100 C. From the pattern of degradation, it is evident that lignin has no effect on the thermal stability of PF. Both LPF and PF degrade similarly. The reduction in residue was related to the amount of lignin replaced in PF.

R: Text explained now the information about Rotary-evaporation of resin, losing 50% mass of volatile compounds (water, methanol), followed by the elimination of free formaldehyde in resin. We cited other curves from other studies. We re-wrote many sections to satisfy you.

  1. Conclusion

Based on your observations, please conclude what is the best composition (optimized parameter) for the LPF.

R: As mentioned in the text, the best composition was 30%, as it presented better mechanical performance in the gluing quality as well as it did not show changes in features, being very similar to traditional PF resin.

Reviewer 3 Report

1- Abstract

The motive of the research is briefly highlighted. Method chosen and findings from the research are also included.

2- Introduction

It is recommended to describe the article's structure in the final sentence of the introduction section.

3- Results

Table 1- What is the conclusion from this table? The solid contents, viscosity, and gel time of PF+10% and PF+50% are significantly different. Which ratio is best, though?

Line 190: What is the minimum standard requirement for the shear test?

4- Conclusions

Line 268: “The physical tests of the four resin formulations indicated that the addition of phenol-lignin tends to decrease the gel time and increase the viscosity”. Kindly check because there are no significant differences between PF and PF+10% in terms of the gel time and viscosity.

Please revised the conclusions and add the limitations and future work of the research.

Author Response

1- Abstract

The motive of the research is briefly highlighted. Method chosen and findings from the research are also included.

R: The introduction was rewritten to evince the goals, above all, about the high purity of the lignin in utilization from Kraft process, being a different raw material when compared to other studies. Furthermore, the synthesis time was reduced to evince the shorter times.

2- Introduction

It is recommended to describe the article's structure in the final sentence of the introduction section.

R: Thank you. We inserted the following paragraph to structure the paper organization: “This paper is organized into the following sections: materials and methods with the respective materials, lignin phenolation, syntheses of the phenol-formaldehyde synthesis and lignin-phenol-formaldehyde, resin properties, mechanical test, thermal test, results on physicochemical and mechanical properties, thermal analyses (differential scanning calorimetry and thermal gravimetric analysis), discussion, and conclusion.”

3- Results

Table 1- What is the conclusion from this table? The solid contents, viscosity, and gel time of PF+10% and PF+50% are significantly different. Which ratio is best, though?

R: Graphs were plotted and inserted to evince the relationship among these properties as you can check now. From regression linear analysis, ideal features were defined for resins.

Line 190: What is the minimum standard requirement for the shear test?

R: As proposed, we carried out the failure analysis using criteria established by the ASTM D-2559(2018), which allowed the conclusion that LPF30 resin may be classified as structural ones, and LPF20 as a non-structural resin. LPF10 and LPF50 did not meet the specifications. Such information was clearly explained in the text.

4- Conclusions

Line 268: “The physical tests of the four resin formulations indicated that the addition of phenol-lignin tends to decrease the gel time and increase the viscosity”. Kindly check because there are no significant differences between PF and PF+10% in terms of the gel time and viscosity.

R: Conclusion was re-written.

Please revised the conclusions and add the limitations and future work of the research.

R: Only LPF30 resin met the structural specifications and, therefore, it was compatible to PF. We concluded that the thermal stability of LPF resins were inferior to PF, suggesting future works to increase the substitution levels and analyse the behavior of resins under fire tests. They were inserted in the final part.

Round 2

Reviewer 1 Report

The authors have done a significant job, the quality of the article has improved

However, some revision of the manuscript is needed.

Table 2: What do the letters "a b c" mean?

Fig 4 - poor quality

Please read the manuscript carefully and make any necessary corrections.

Author Response

The authors have done a significant job, the quality of the article has improved.

R: We appreciate your efforts to support us in this revision process. Significant changes were done to improve the quality of our paper to satisfy your suggestions and remarks.

However, some revision of the manuscript is needed.

R: We appreciate your remarks, and we declared that we tried to satisfy them.

Table 2: What do the letters "a b c" mean?

R: We inserted it in the legend.

Fig 4 - poor quality

R: We appreciate your observation. We changed this image by other with a finer quality.

Please read the manuscript carefully and make any necessary corrections.

R: We read and check carefully the paper.

The revised paper is inserted for your final appreciation. Thank you.

Reviewer 2 Report

The authors have made significant changes, which, in my opinion, are sufficient for their work to be accepted for publication in Polymers.

Author Response

The authors have made significant changes, which, in my opinion, are sufficient for their work to be accepted for publication in Polymers.

R: We appreciate your efforts to support us in this revision process. Significant changes were done to improve the quality of our paper to satisfy your suggestions and remarks.

We are sending our version with final suggestions of other reviewer (changes in red). Thank you.
